

# Impact of intercropping on the coupling between soil microbial community structure, activity, and nutrient-use efficiencies

Tengxiang Lian[1,2], Yinghui Mu[1,2], Jian Jin[3], Qibin Ma[1,2], Yanbo Cheng[1,2], Zhandong Cai[1,2] and Hai Nian[1,2]

[1] The State Key Laboratory for Conservation and Utilization of Subtropical Agro-bioresources, South China Agricultural University, Guangzhou, China
[2] The Key Laboratory of Plant Molecular Breeding of Guangdong Province, College of Agriculture, South China Agricultural University, Guangzhou, China
[3] Key Laboratory of Mollisols Agroecology, Northeast Institute of Geography and Agroecology, Chinese Academy of Sciences, Harbin, Heilongjiang, China

Corresponding author
Hai Nian, hnian@scau.edu.cn

## ABSTRACT

Sugarcane-soybean intercropping has been widely used to control disease and improve nutrition in the field. However, the response of the soil microbial community diversity and structure to intercropping is not well understood. Since microbial diversity corresponds to soil quality and plant health, a pot experiment was conducted with sugarcane intercropped with soybean. Rhizosphere soil was collected 40 days after sowing, and MiSeq sequencing was utilized to analyze the soil microbial community diversity and composition. Soil columns were used to assess the influence of intercropping on soil microbial activity (soil respiration and carbon-use efficiency: nitrogen-use efficiency ratio). PICRUSt and FUNGuild analysis were conducted to predict microbial functional profiling. Our results showed that intercropping decreased pH by approximately 8.9% and enhanced the soil organic carbon, dissolved organic carbon, and available nitrogen (N) by 5.5%, 13.4%, and 10.0%, respectively. These changes in physicochemical properties corresponded to increased microbial diversity and shifts in soil microbial communities. Microbial community correlated significantly ($p < 0.05$) with soil respiration rates and nutrient use efficiency. Furthermore, intercropping influenced microbial functions, such as carbon fixation pathways in prokaryotes, citrate cycle (TCA cycle) of bacteria and wood saprotrophs of fungi. These overrepresented functions might accelerate nutrient conversion and control phytopathogens in soil.

## INTRODUCTION

Sugarcane-soybean intercropping has been widely used to stabilize yields and reduce nitrogen leaching (*Rowe et al., 2005*; *Xu, Li & Shan, 2008*; *Li et al., 2013*). N fixation associated with soybeans can improve soil fertility and field ecological conditions

that favor sugarcane in the intercropping system (*He et al., 2006*). Intercropping of sugarcane with soybean, may also stimulate N fixation by the legume's microbiome (*Li et al., 2013*).

In an intercropping system, the roots of different plant species interact directly with each other and subsequently affect root exudation, which undoubtedly alters the microbial diversity, structure, and activity (*Zhou, Yu & Wu, 2011*; *Broeckling et al., 2008*; *Gomes et al., 2003*). The changed microbial community and activity by intercropping could affect C and N dynamics (*Kaur, Gupta & Singh, 2000*; *Rowe et al., 2005*; *Sun et al., 2009*), and this may be attributed to the ability of microbial communities to regulate carbon and nitrogen-use efficiency (*Mooshammer et al., 2014*). Thus, a comprehensive method that incorporates the carbon-use efficiency: nitrogen-use efficiency ratio and soil respiration could be used to evaluate the change in microbial activity caused by the microbial community (*Zhong, Yan & Shangguan, 2015*).

The influence of intercropping on the soil microbial communities have been studied in several intercropping systems, including mulberry–soybean, *Eucalyptus–Acacia mangium* and apple tree-crown vetch intercropping (*Li et al., 2013*, *2016*; *Rachid et al., 2015*; *Zheng et al., 2018*). For example, *Li et al. (2016)* investigated the effects of mulberry–soybean intercropping on the diversity and composition of the soil bacterial community in salt–alkali soil and found that intercropping increased the abundance of some phosphate solubilizing species. Moreover, *Rachid et al. (2015)* reported that *Eucalyptus* intercropped with *Acacia mangium* increased soil fungal community diversity and changed the fungal structure and *Zheng et al. (2018)* reported that intercropping of apple trees and crown vetch changed the soil bacterial structure but not diversity. In our study, the bacterial and fungal structure and activity in the intercropping and monoculture system were analyzed. We hypothesized that intercropping improves soil properties, increases the microbial diversity, changes community structure and improves some microbial function (H1) and that change in microbial community will correlate with microbial activity (H2).

## MATERIALS AND METHODS

### Experimental design and plant materials

The intercropping experiment was established in March of 2016 with three replicates of three treatments in a randomized block design. The treatments included (1) sugarcane monoculture, (2) soybean monoculture, and (3) sugarcane intercropped with soybean. The soil used in this study was classified as Ali-Udic Argosol with pH 5.1, soil organic carbon (SOC) 8.5 g kg$^{-1}$, 0.41 g kg$^{-1}$ total N, and 0.42 g kg$^{-1}$ total P.

The sugarcane variety ROC22 (*Saccharum officinarum*) and soybean variety HuaChun5 (*Glycine max L.*), which are widely grown in South China, were used in this study. Plants were grown in pots in the glasshouse at South China Agriculture University, Guangzhou, China. In brief, all plants within a pot (140 cm wide × 45 cm width × 45 mm high) were filled with 30 kg of sieved soil (<2 mm) and considered as one replicate. Two sugarcane seedlings or three soybean seeds were planted in a pot under the monoculture system, or two sugarcane seedlings with three soybean seeds were planted

under the intercropping system. The row space was 0.9 m for sugarcane and 0.3 m for soybean in all treatments. The water content of the soil was adjusted to 80% of field water capacity. Plants were harvested at the flowering stage.

## Soil sampling and measurements

Rhizosphere soil was recovered separately on 25 May 2016 (40 days after sowing) by shaking root for 3 min into a bag and mix thoroughly. Contact between samples was avoided. Approximately five g soil from each treatment was collected and stored at −80 °C for DNA extraction. Additionally, 100 g soil was collected and stored at 4 °C for analyses of microbial and soil physicochemical properties.

Soil pH was determined in a soil-water slurry (1:5 w:v) using a pH meter (FE20-FiveEasy™ pH; Mettler Toledo, Weilheim, German). Soil total nitrogen was measured using an elemental analyser (VarioEL III; Elementar Analysensysteme GmbH, Frankfurt, Germany). Nitrate ($NO_3^-$) and ammonium ($NH_4^+$) were assayed using a continuous flow analytical system (SKALAR SAN++; Skalar Inc., Breda, The Netherlands). SOC, dissolved organic carbon (DOC), and dissolved organic nitrogen (DON) were measured using a TOC analyzer (Multi N/C 2100; Analytik Jena, Jena, Germany). Soil microbial biomass carbon (MBC) and microbial biomass nitrogen (MBN) were measured by the chloroform-fumigation extraction method (*Vance, Brookes & Jenkinson, 1987*). The sum value of $NO_3^-$, $NH_4^+$, and DON was considered as available nitrogen (available N).

## Soil incubation and respiration measurements

The methods of the incubation experiment were reported in our previous study (*Lian et al., 2016*). In brief, 20 g soil was collected from each treatment and placed into PVC cores (five cm height, 2.5 cm diameter). The PVC core and a beaker with 10 mL 1M NaOH, which used to trap $CO_2$, were placed into a 0.5 L sealed container.
The trapped $CO_2$ was precipitated with 0.5M $SrCl_2$ and NaOH was neutralized with 0.1M HCl. Soil respiration was estimated on 1, 3, 5, 7, 9, 11, 14, 18, 22, 26, 32, 39, 46, 53, and 60 days after incubation was initiated (*Blagodatskaya et al., 2011*).

## DNA extraction and quantitative PCR (qPCR)

DNA was extracted using Fast DNA SPIN Kit for Soil (Qbiogene Inc., Carlsbad, CA, USA) according to the manufacturer's instructions. Quantitative PCR was conducted by targeting bacterial 16S rRNA genes and fungal ITS1 region, using the primers 515F/907R (*Osburn et al., 2011*) and ITS1F/ITS2R (*Yao et al., 2017*), respectively, following the protocols reported previously (*Liu et al., 2015*).

## Illumina MiSeq sequencing analysis

Illumina MiSeq sequencing of the 16S rRNA genes and fungal ITS1 region was performed to examine the structure of the soil bacterial and fungal community, respectively. The raw sequences were processed and analyzed using QIIME1 Pipeline Version 1.9.0.
Multiple steps were conducted to remove low-quality sequences with lengths shorter than 200 bp and quality scores less than 20. For further analysis, the chimeric sequences were checked and removed using UCHIME algorithm. High-quality sequences were

clustered into operational taxonomic units (OTUs) using Ribosomal Database Project (RDP) Classifier based on 97% sequence similarity. The OTUs were analyzed using the SILVA and UNITE database for bacteria and fungi, respectively. Then, a phylogenetic tree was built using Fast Tree (*Price, Dehal & Arkin, 2009*). For a correct comparison between samples, rarefied subsequenting numbers (14,811 for bacterial and 29,726 for fungi) were used for subsequent analysis. All sequences have been deposited into the GenBank short-read archive under accession SRP116883 (bacteria) and SRP129902 (fungi).

### Statistical analysis

Using the program R (*R Core Team, 2018*; vegan package), principal coordinate analysis (PCoA) based on OTU level was processed to assess the patterns of similarity (Bray–Curtis similarity) in the composition of the microbial community between treatments. The Chao 1 index and Shannon richness were calculated to compare soil bacterial and fungal alpha diversity (Fig. S1). A canonical correspondence analysis (CCA) was conducted to reveal the association between soil property variables and microbial community structure. Spearman correlation analysis was conducted with SPSS 24.0 to identify correlation between microbial activity and structure. PICRUSt analysis and STAMP were conducted to predict and visualize bacterial functional profiling (*Langille et al., 2013*; *Parks & Beiko, 2010*). FUNGuild was used to identify fungi functional guilds (*Nguyen et al., 2016*).

ANOVA test was used with Genstat 13 (VSN International, Hemel Hemspstead, UK) to assess the effect of treatments on the SOC, total N, MBC, MBN, DOC, DON, $NH_4^+$-N, $NO_3^-$ -N, pH, and the relative abundance of OTU inferred with FUNGuild. Furthermore, ANOVA test of least significant difference was used to assess the different of respiration rate and cumulative respiration. Differences were considered statistically significant at level of $p < 0.05$. The ratios of microbial community carbon-use efficiency and nitrogen-use efficiency were calculated as follows (*Mooshammer et al., 2014*):

### Carbon-use efficiency: Nitrogen-use efficiency = $B_{C:N}$:$R_{C:N}$

Where $B_{C:N}$ is the C:N ratio of the microbial community and $R_{C:N}$ is the C:N ratio of the soil.

## RESULTS

### Effect of intercropping on soil physicochemical properties

Compared with monoculture treatments, intercropping decreased pH from 6.73 to 6.13 and from 5.97 to 5.45 for sugarcane and soybean, respectively. The SOC was higher in intercropped sugarcane (I-Sugarcane) and intercropped soybean (I-Soybean) than that in the sugarcane monoculture (M-Sugarcane) and soybean monoculture (M-Soybean). The concentration of SOC significantly increased under I-Sugarcane compared with that under M-Sugarcane. $NH_4^+$, DOC, DON, MBC, and MBN levels were significantly increased for the two plant species under intercropping compared with those for the monoculture ($p < 0.05$), while the $NO_3^-$ level showed an opposite trend ($p < 0.05$) (Table 1).

**Table 1 Soil physiochemical properties at different culture mode.**

| | pH | SOC (g kg$^{-1}$) | TN (g kg$^{-1}$) | NH$_4^+$-N (mg kg$^{-1}$) | NO$_3^-$-N (mg kg$^{-1}$) | DOC (mg kg$^{-1}$) | DON (mg kg$^{-1}$) | MBC (mg kg$^{-1}$) | MBN (mg kg$^{-1}$) | Available N (mg kg$^{-1}$) |
|---|---|---|---|---|---|---|---|---|---|---|
| M-Sugarcane | 6.73 ± 0.06a | 16.25 ± 0.28c | 0.73 ± 0.06b | 5.66 ± 066c | 2.02 ± 0.19a | 97.67 ± 2.51b | 0.91 ± 0.01b | 131 ± 21.66b | 6.40 ± 0.57ab | 8.58 ± 0.48b |
| I-Sugarcane | 6.13 ± 0.15b | 17.15 ± 0.21a | 0.81 ± 0.01a | 8.10 ± 0.25b | 0.42 ± 0.10d | 109 ± 3.61a | 0.79 ± 0.03b | 198 ± 13.08a | 8.12 ± 0.51a | 9.65 ± 0.08b |
| M-Soybean | 5.97 ± 0.21b | 16.30 ± 0.57bc | 0.81 ± 0.02a | 6.68 ± 0.87bc | 2.00 ± 0.29b | 90.33 ± 1.52c | 0.62 ± 0.10c | 133 ± 26.46b | 5.60 ± 0.06c | 12.30 ± 0.58a |
| I-Soybean | 5.45 ± 0.06c | 16.93 ± 0.08ab | 0.77 ± 0.03ab | 10.35 ± 1.18a | 1.17 ± 0.18c | 106 ± 3.61ba | 1.10 ± 0.14a | 234 ± 27.62a | 6.69 ± 0.40a | 12.62 ± 0.51a |

Notes:
The soil physiochemical properties and microbial biomass carbon (MBC) and nitrogen (MBN) in each soil are shown in the table. Values are the means ± SE ($n = 3$). M-Sugarcane, sugarcane monoculture; I-Sugarcane, intercropped sugarcane. M-Soybean, soybean monoculture; I-Soybean, intercropped soybean.

## Effect of intercropping on microbial activity

Soil respiration was enhanced in the intercropping system (Fig. 1). Respiration peaked on day one and then decreased exponentially; respiration did not differ significantly between treatments (Fig. 1A). During the incubation, the cumulative CO$_2$-C levels were 6.9% and 5.3% greater in the intercropping soil than those in the monoculture for sugarcane and soybean, respectively (Fig. 1B). Moreover, the I-Sugarcane and I-Soybean treatments showed higher ratios of carbon- and nitrogen-use efficiency than that for the M-Sugarcane and M-Soybean, respectively (Fig. 1C).

Intercropping significantly increased bacterial and fungal abundances in sugarcane. However, no significant difference was observed in the fungi:bacteria (F:B) ratios (Fig. 2). Moreover, intercropping increased Chao index of bacteria and fungi; however, significant higher Shannon richness was only observed in I-Soybean compared with M-Soybean (Fig. S1D). PCoA analysis showed that the bacterial and fungal communities from different treatments were clearly separated from each other (Fig. S2), which indicates crop species and culture mode influenced the soil microbial community. The dominant phyla, Proteobacteria, Chloroflexi, Acidobacteria, Actinobacteria, and Firmicutes for bacteria and Ascomycota, Zygomycota and Basidiomycota for fungi, were the same across treatments (Fig. 3).

Canonical correspondence analysis (Fig. 4) revealed a relationship between microbial community structure and soil property variables. Significance values for the overall solution and for the CCA1 and CCA2 axes were 0.035, 0.041, and 0.039 for the bacterial community and 0.005, 0.01, and 0.005 for the fungal community, respectively. The soil pH, SOC, NO$_3^-$, DOC, MBC, and MBN ($p = 0.04, 0.01, 0.03, 0.008, 0.02$, and $0.004$, respectively, for the bacterial community; $p = 0.001, 0.04, 0.04, 0.01, 0.003$, and $0.02$ for the fungal community) appeared to be strongly correlated with the microbial community.

The X-axis of the PCoA analysis of bacteria and fungi (Fig. S2) was used in the Spearman correlation analysis to detect the relationships between microbial activity and structure (Table 2). A significant relationship was observed between bacterial community and microbial activity; however, the fungal community was not significantly correlated with bacterial communities and microbial activity.

## Effect of intercropping on microbial functional characteristics

Bacterial function predictions were categorized into KEEG pathways. In brief, pathways for nutrient cycles such as carbon fixation pathways in prokaryotes and citrate cycle

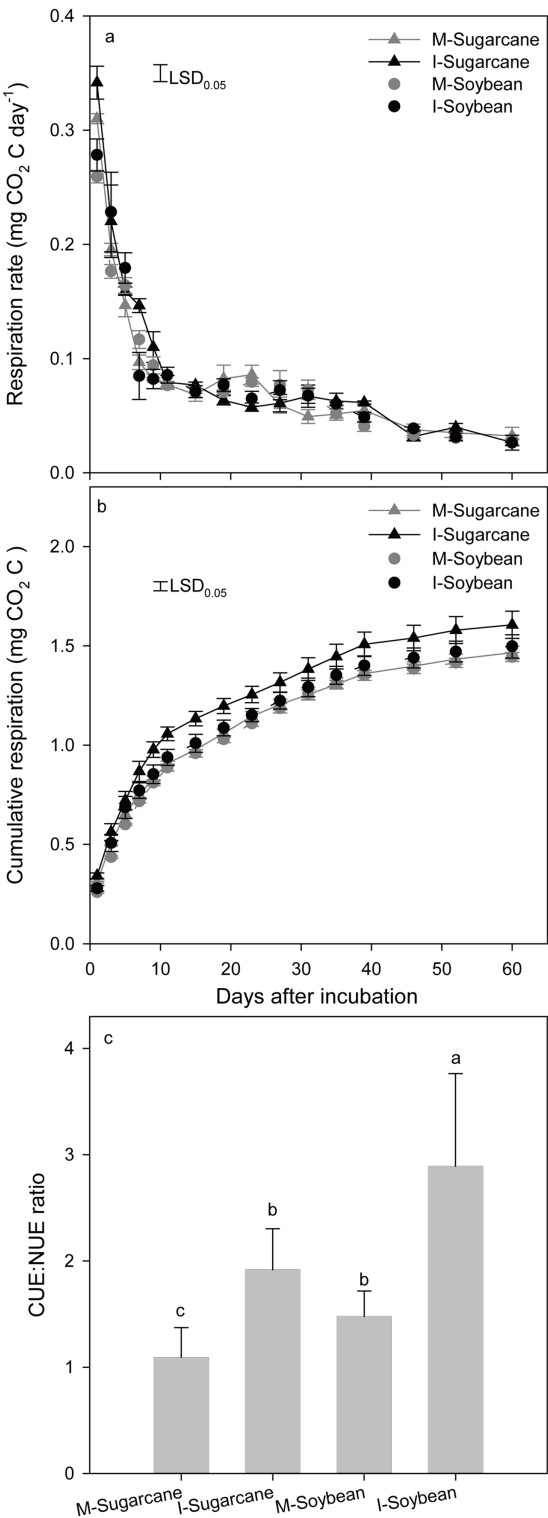

**Figure 1 Respiration rates, cumulative respiration and the ratios of carbon-use efficiency and nitrogen-use efficiency.** Respiration rates (A) and cumulative respiration (B) each treatment and the ratios of carbon-use efficiency and nitrogen-use efficiency (C). M-Sugarcane, sugarcane monoculture; I-Sugarcane, intercropped sugarcane. M-Soybean, soybean monoculture; I-Soybean, intercropped soybean.

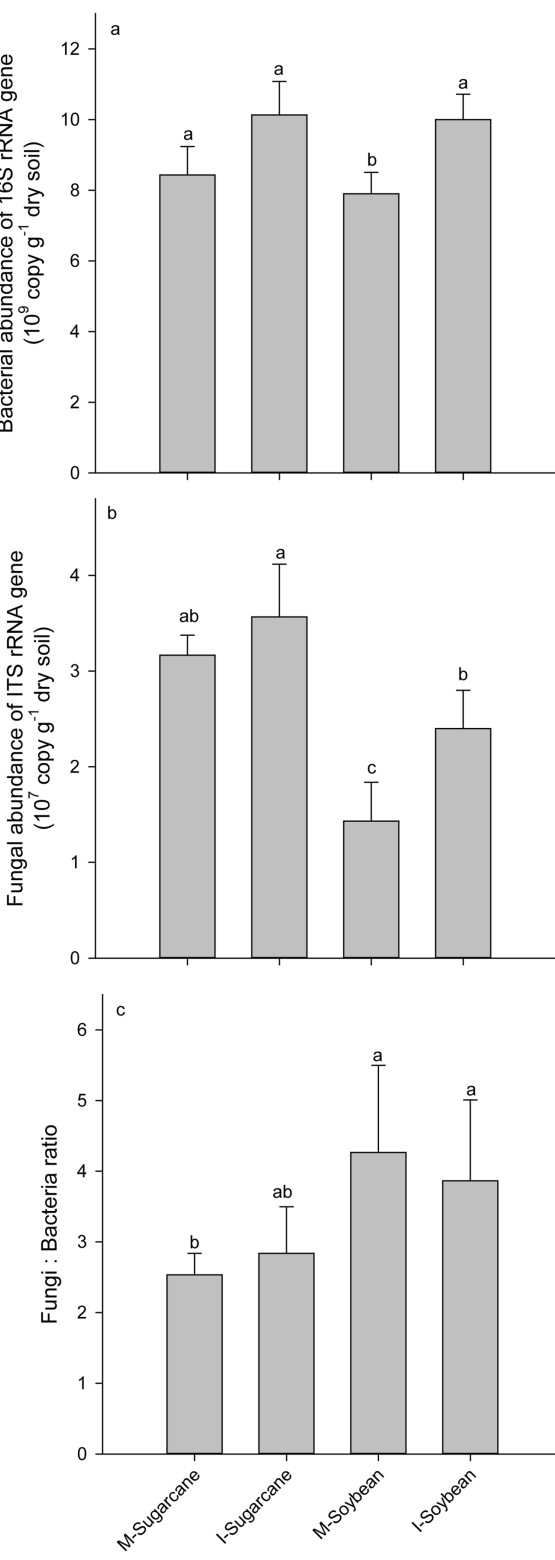

**Figure 2 Bacterial and fungal abundances, and the relative ratio of the fungi and bacteria.** Bacterial abundances (A) and fungal abundances (B) and relative ratio of the fungi and bacteria (C) in each treatment. M-Sugarcane, sugarcane monoculture; I-Sugarcane, intercropped sugarcane. M-Soybena, soybean monoculture; I-Soybean, intercropped soybean.

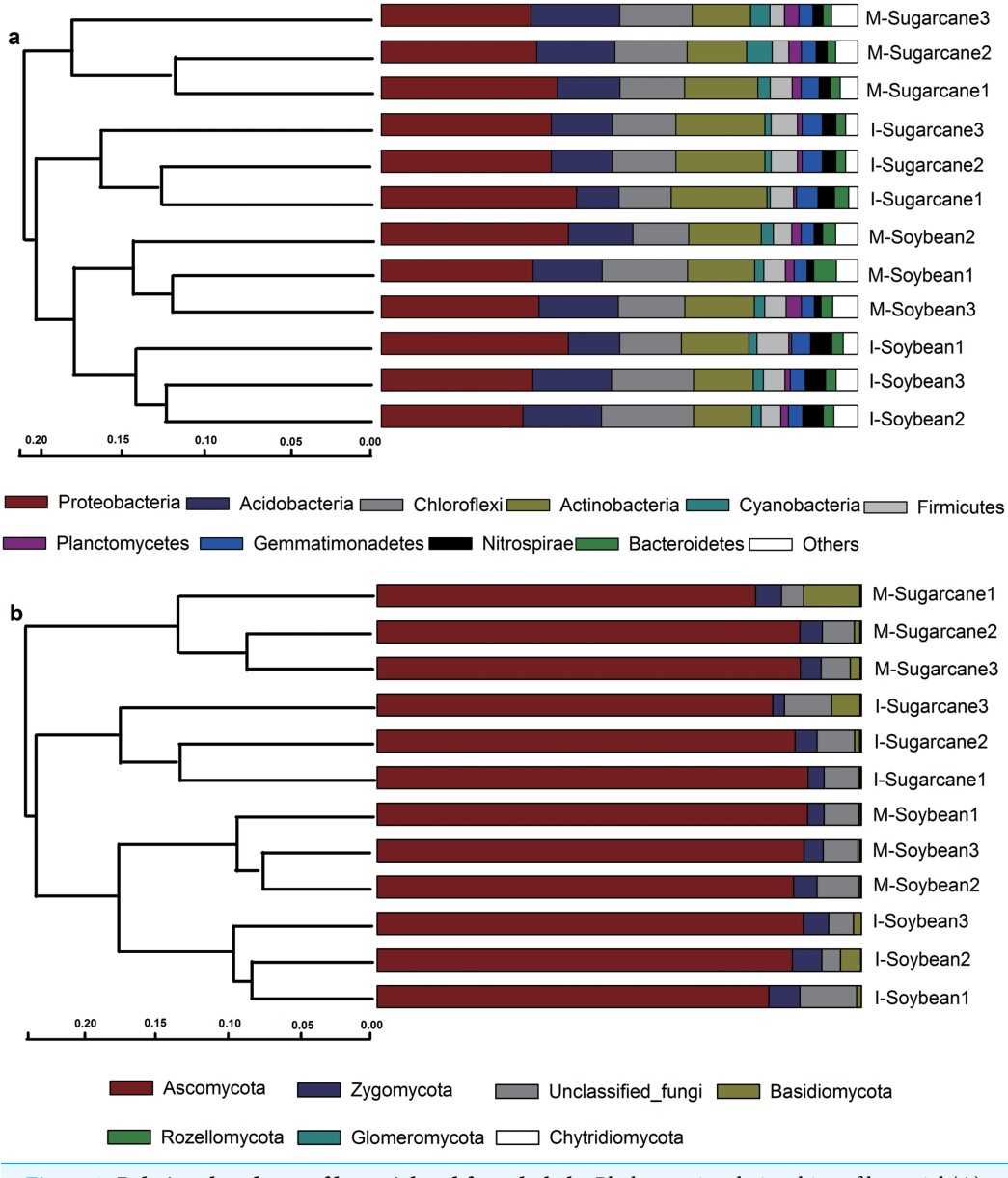

**Figure 3** **Relative abundance of bacterial and fungal phyla.** Phylogenetic relationships of bacterial (A) and fungal (B) communities shown with the relative abundances of different phyla. M-Sugarcane, sugarcane monoculture; I-Sugarcane, intercropped sugarcane. M-Soybean, soybean monoculture; I-Soybean, intercropped soybean.

(TCA cycle) significant increased, while lipid metabolism, sulfur metabolism and signal transduction mechanisms significant decreased in both I-Sugarcane and I-Soybean (Fig. 5; Table S1). For fungal function, 12 fungal functional guilds, such as wood saprotroph, plant pathogen, plant saprotroph, fungal parasite and endophyte were detected (Fig. 6A). Among the top five fungal functional guilds, the relative abundances of wood saprotroph significant increased in I-Sugarcane, and plant pathogen significant increased in I-Soybean (Table S2).

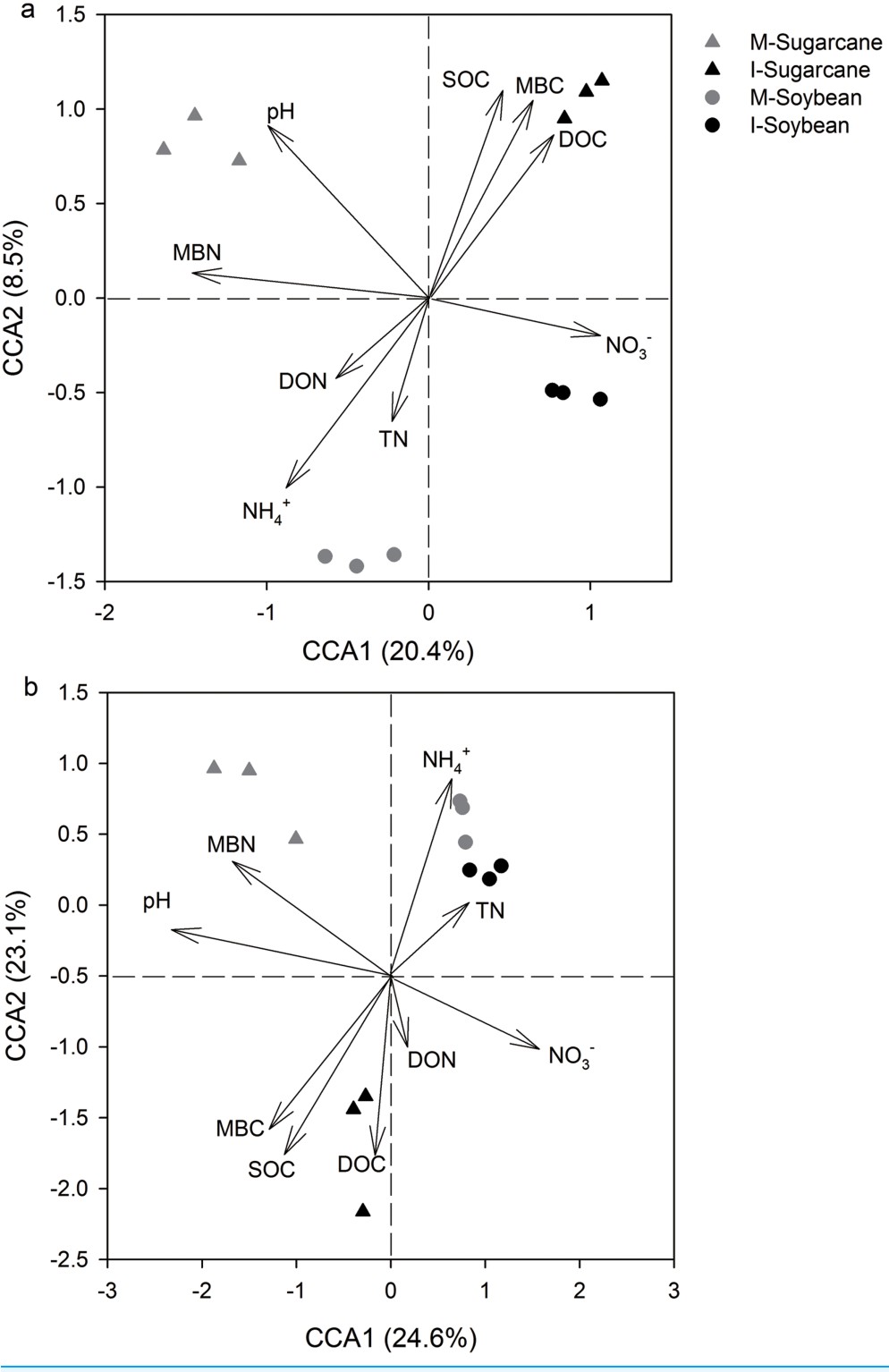

**Figure 4 Canonical correspondence analysis (CCA) of bacterial and fungal communities.** Canonical correspondence analysis (CCA) of bacterial and fungal communities Canonical correspondence analysis (CCA) of bacterial (A) and fungal (B) communities changes with environmental variables M-Sugarcane, sugarcane monoculture; I-Sugarcane, intercropped sugarcane. M-Soybean, soybean monoculture; I-Soybean, intercropped soybean.

**Table 2  Spearman's correlations between microbial community structure and microbial activity.**

| | Fungal community | Ratio of carbon- and nitrogen-use efficiency | Soil respirations |
|---|---|---|---|
| Bacterial community | −0.034 | 0.787** | 0.609* |
| Fungal community | | −0.392 | 0.537 |

Notes:
*$P < 0.05$.
**$P < 0.01$ indicate significant correlations.

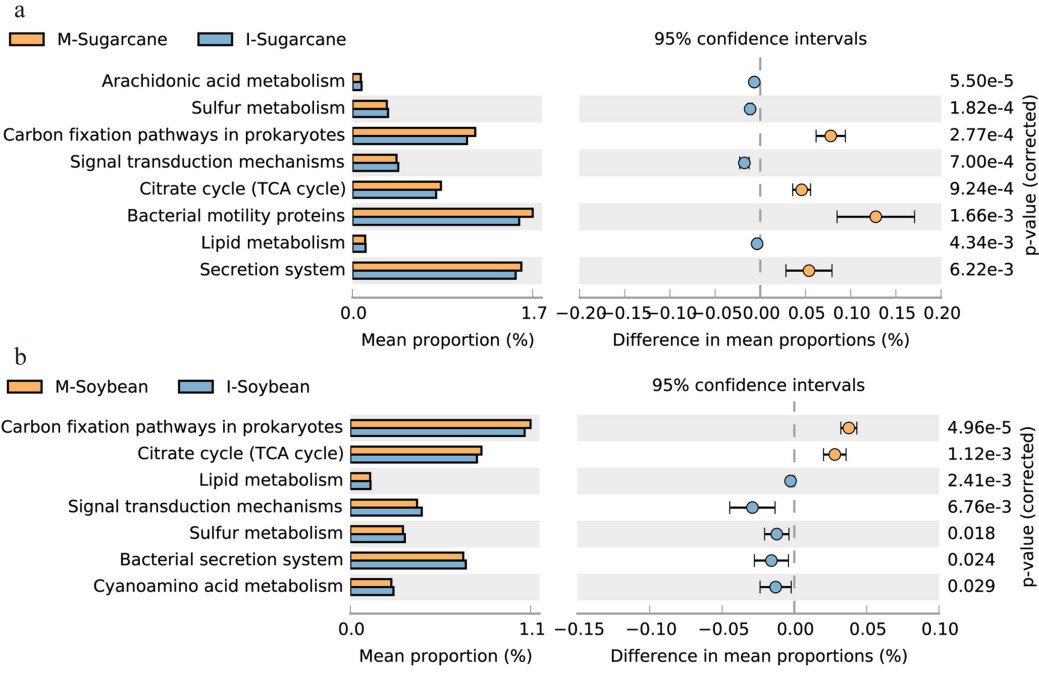

**Figure 5  Relative abundance of predicted soil bacterial functions.** Relative abundance of predicted soil bacterial functions in sugarcane (A) and soybean (B) system predicted by PICRUSt using KEGG Orthologs. Pathways presented here are relevant to soil ecosystem function, full data supplied in Table S1. I-Sugarcane, intercropped sugarcane. M-Soybean, soybean monoculture; I-Soybean, intercropped soybean.

From the library generated with the ITS primer pair, more than 40 OTUs were detected and assigned to the functional group of wood saprotrophs. The top 4 OTUs belonged to the phylum Ascomycota, with relative abundance ranged from 6.21% to 18.38% (Table S3). Of these, the relative abundances of OTU133 (*Trichoderma*) in I-Sugarcane were significantly higher than M-Sugarcane. Relative abundances of OTU1092 (*Aspergillus*) and OUT126 (*Acremonium*) were significantly higher in both I-Sugarcane and I-Soybean (Fig. 6B). For the functional group of plant pathogen, the top 4 OTUs were belonged to the phylum Ascomycota. Among them, relative abundances of OTU745 (*Gibberella*), OTU1092 (*Clonostachys*), and OUT126 (*Gibellulopsis*) were general lower in intercropping system. Nevertheless, the relative abundance of OTU941 (*Gibellulopsis*) was significantly higher in both I-Sugarcane and I-Soybean (Fig. 6C; Table S4).

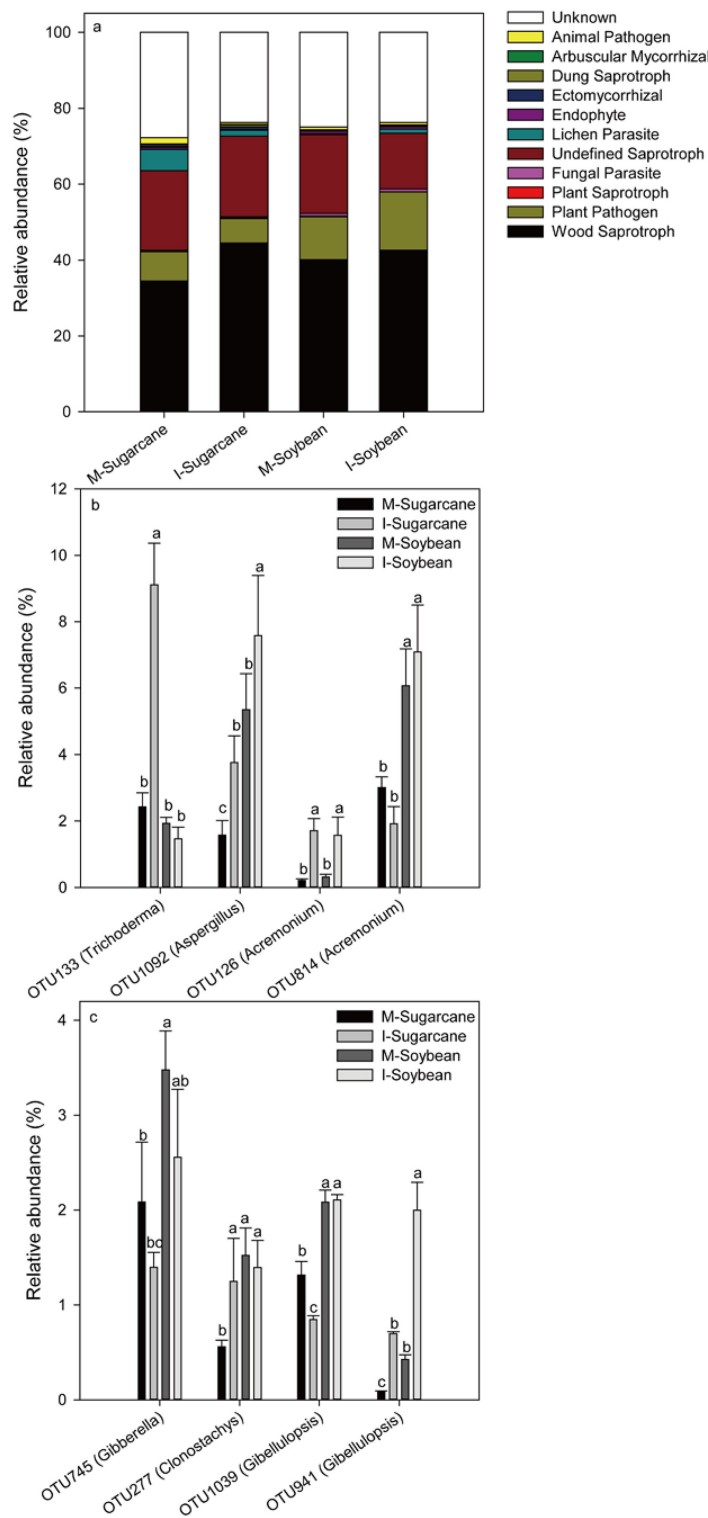

**Figure 6 Relative abundance of predicted soil fungal functions.** Relative abundance of predicted soil fungal functions (A) and relative abundance of fungal functional OTUs inferred by FUNGuild belong to wood saprotrophs (B) and plant pathogens (C). Full data supplied in Table S2–S4. I-Sugarcane, inter-cropped sugarcane. M-Soybean, soybean monoculture; I-Soybean, intercropped soybean.

## DISCUSSION

### Effect of intercropping on soil microbial communities

Intercropping of sugarcane with soybean improved microbial respiration, bacterial and fungal abundances, and diversity, which is consistent with several previous studies (*Sun et al., 2009*; *Kaur, Gupta & Singh, 2000*; *Li et al., 2016*). For soils collected from the intercropping or monoculture systems, nutrients and respiration decreased markedly from initial rates, which agrees with earlier studies (*Lian et al., 2016*) and the improvements in soil quality associated with intercropping appeared temporary.
The higher respiration rates in the intercropped system could relate to nitrogen fixation by the legume, which may have stimulated soil respiration of the intercropped sugarcane (*Qin et al., 2013*). Higher bacterial and fungal abundances and diversity may reflect the direct contact of crop roots in the intercropping system which stimulates roots to release more nutrients (*Song et al., 2007*). Environmental factors, such as pH and SOC, often play important roles in shaping microbial community composition and diversity (*Hartman et al., 2008*; *Tripathi et al., 2018*; *Ma et al., 2018*). Soil pH influences the acid–base equilibrium of microbial cells or regulates the availability of soil nutrients (*Zhalnina et al., 2015*). In this study, soil pH in intercropped soybean soils decreased compared with that in monoculture soil. Additionally, CCA showed pH strongly correlated with ($p = 0.04$ and $p = 0.001$, for the bacterial and fungal community, respectively), the microbial community. Together, our data indicates that pH governed the microbial community.

Soil organic carbon can also determine microbial community structure in natural environments (*Sul et al., 2013*; *Ma et al., 2018*), as increased SOC favors copiotrophes (*Fierer et al., 2012*; *Waring, Averill & Hawkes, 2013*; *Luo et al., 2018*). In this study, the significant increased SOC in intercropping system and the significant correlation ($p = 0.01$ and $p = 0.04$, for the bacterial and fungal community, respectively), between SOC and microbial community (Table 1; Fig. 4) indicate that SOC shaped the structure of microbial community. $NO_3^-$, DOC, MBC, and MBN may also have significant impact on the shift of microbial community structure (Fig. 4). Shifting nutrients changes microbial communities (*Sun et al., 2015*; *Ramirez et al., 2010*), since increases in available substrates might increase the activity of copiotrophs in soil (*Fierer et al., 2012*).

### Effect of intercropping on microbial activity

Carbon-use efficiency, which can be calculated as $C_{growth}/(C_{growth} + C_{respiration})$ has a positive correlation with microorganism growth and a negative correlation with respiration rates (*Spohn et al., 2016*). The increased microbial abundance, without a change in available N, may explain the increase in carbon-use efficiency and not much change in nitrogen-use efficiency. Therefore, the increased ratios of carbon- and nitrogen-use efficiency in this study may be caused by carbon-use efficiency. However, respiration and the ratios of carbon- and nitrogen-use efficiency, which represent the microbial activity, both increased in the intercropping system in this research. This may be correlated with the nonlinear relationship between microbial growth and respiration (*Sinsabaugh et al., 2013*).
There have many other factors that affect the carbon-use efficiency such as environmental constraints and resource (*Sinsabaugh et al., 2013*).

Carbon-use efficiency is positively correlated with nutrient availability. Bacterial carbon-use efficiency tends to increase more markedly with nutrient availability than that with fungi (*Keiblinger et al., 2010*), as bacteria and fungi are usually considered copiotrophic (r-selected) and oligotrophic (K-selected) groups (*Geyer et al., 2016*). Additionally, the ratio of fungi and bacteria might shift soil microbial community carbon-use efficiency (*Sinsabaugh et al., 2013*). In our research, the abundance of bacteria and fungi were significant higher in the I-Soybean system. However, the fungi:bacteria ratio did not differ among treatments. Together, more bacteria and fungi abundance and the improved nutrient availability, such as SOC and DOC, in intercropping system may lead to an increase carbon-use efficiency, which subsequently increased the ratios of carbon-use efficiency and nitrogen-use efficiency. Additionally, better nutrient availability in the plant-soil-microbial system could improve the plant growth. The subsequent release of exudates increases soil microbial activity (*Zhong, Yan & Shangguan, 2015*; *Levy et al., 2018*).

## The relationship between microbial activity and microbial structure

The changed microbial community structure may result in increased microbial activity in the intercropping system. Analyzing dozens of species soil samples from a wide range of ecosystems across America, *Fierer, Bradford & Jackson (2007)* generalized that C mineralization predicts bacterial phyla abundances. Thus, we assumed that the changed microbial activity may be caused by the shift in the structure of microorganisms. The correlation between the microbial activity and community structure showed that the bacterial community was significantly correlated with ratios of carbon- and nitrogen-use efficiency and respiration. However, this significant correlation was not observed with the fungal community, which may be attributed to bacteria were 100 times more abundance than fungi (Fig. 2). Therefore, compared to bacteria, the changed fungal community might have a less effect on microbial activity. Nevertheless, the dormancy rates of bacteria are generally higher than fungi (*Jones & Lennon, 2010*). When the dormancy rate reaches a certain level, the quantitative advantage of bacteria cannot explain the high microbial activity and this requires further verification and research. In this study, pH, $NO_3^-$, MBN, and SOC play important roles in the change in community structure. Our findings indicate that intercropping altered the availability of carbon in the soil. The changed nutrient subsequently allowed bacteria to affect the carbon- and nitrogen-use efficiency of the microorganisms.

## Effect of intercropping on soil functional microorganisms

In general, the prediction of functional microorganisms matched well with our expectations. Functions such as carbon fixation pathways in prokaryotes, citrate cycle (TCA cycle) of bacteria and wood saprotrophs of fungi were higher in the intercropping system. In contrast, plant pathogens were slightly lower in the intercropping system. The result indicated that the overrepresented functions in intercropping system potentially

leading to an accumulation of metabolic products and nutrients. For example, the increased carbon fixation pathways in prokaryotes and citrate cycle (TCA cycle) suggested an acceleration of nutrient conversion (*Shi et al., 2017*), which might be trigged by the increased microbial activity. Furthermore, the increased OTU133 (*Trichoderma*) belong to wood saprotrophs in intercropped sugarcane may control a wide range of phytopathogens because *Trichoderma* secretes chitinases and cellulases, which can hydrolyse pathogen cell walls (*Bae et al., 2015*).

## CONCLUSIONS

In conclusion, sugarcane-soybean intercropping in acidic soil increased microbial diversity and shifted soil microbial communities and soil physicochemical properties (pH, SOC, $NO_3^-$, DOC, MBC, and MBN). The changed bacterial community correlated to higher soil respiration rates and nutrient use efficiency. Furthermore, intercropping influenced microbial functions, such as carbon fixation pathways in prokaryotes, citrate cycle (TCA cycle) of bacteria and wood saprotrophs of fungi. These overrepresented functions may accelerate nutrient conversion and control phytopathogens in soil.

### Funding

This work was supported by the National Natural Science Foundation of China (31700091) and the National Key R&D Program of China (2017YFD0101500). The funders had no role in study design, data collection and analysis, decision to publish, or preparation of the manuscript.

### Grant Disclosures

The following grant information was disclosed by the authors:
National Natural Science Foundation of China: 31700091.
National Key R&D Program of China: 2017YFD0101500.

### Competing Interests

The authors declare that they have no competing interests.

### Author Contributions

- Tengxiang Lian conceived and designed the experiments, performed the experiments, analyzed the data, contributed reagents/materials/analysis tools, prepared figures and/or tables, authored or reviewed drafts of the paper, approved the final draft.
- Yinghui Mu conceived and designed the experiments, performed the experiments, contributed reagents/materials/analysis tools.
- Jian Jin authored or reviewed drafts of the paper.
- Qibin Ma analyzed the data.
- Yanbo Cheng performed the experiments.
- Zhandong Cai performed the experiments.

- Hai Nian conceived and designed the experiments, contributed reagents/materials/ analysis tools, authored or reviewed drafts of the paper, approved the final draft.

## Data Availability

Soil samples of the sugarcane soybean intercropping system Illumina MiSeq are provided in the Supplemental Materials. The root system of fungi in soil samples were deposited in GenBank Sequence Read Archive (SRA) with accession numbers SRP116883 and SRP129902.

## Supplemental Information

Supplemental information for this article can be found online at http://dx.doi.org/10.7717/peerj.6412#supplemental-information.

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
