# Peer review of "Impact of intercropping on the coupling between soil microbial community structure, activity, and nutrient-use efficiencies"

_PeerJ, doi:10.7717/peerj.6412_

## Round 0.1 · original submission · Major Revisions

The reviewers found the work interesting in general but had questions about the scientific content that must be addressed in a resubmission. They also suggested revisions for style, which should be addressed. I add to those comments (see below).

Regards,

Michael

Line 23. It is awkward to use the word “response” twice in the sentence and the text is wordy. Revise to “..and structure to intercropping is not well understood. Since microbial diversity corresponds to soil quality and plant health, a pot experiment..”
Line 25. Delete article “the”
Line 27. Avoid the adverb moreover, as it adds little. Revise to “…compostion. Soil columns were used to assess the influence of intercropping on soil microbial activity.”
Line 29. Delete “subsequently effected”
Line 30 – 32. Present results in the order of methods (micobial community then activity) also, the abstract should present nutrient use efficiency results. The title does.
Lines 37. Delete phrase “in the world” and revise to “..used to stabilize yields and reduce nitrogen leaching”
Line 39. Here and throughout, avoid phrases like “Studies have shown that,”
“Many studies have reported (line 47)” or “previous studies have shown (line 54)”
Line 41. Revise to “Intercroping of sugarcane with soybean, may also stimulate N fixation by the legume’s microbiome.”
Line 62. Delete “has”
Line 63. Italicize genus and species.
Line 66. Delete “The aim of this study,” and revise to “…analyzed to (1) explore..”
Line 72. “..experiment was established in ___(month) of 2016 with three replicates of three treatments in a randomized block design”
Line 96. Be consistent in sentence structure. Revise to “MBC and MBN were measured..”
Line 143. Start paragraphs with a topic sentence and not a statement that directs the reader elsewhere (in this case to figure 1).
Line 156. Present results in past tense. Revise to “…revealed a relationship between..”
Line 170. Avoid the passive voice. Revise to “Intercropping ..improved microbial respiration…”
Line 223. Phrase “to improve plant growth” adds little.

Reviewer 1 ·

Basic reporting

Overall, it is an interesting topic to investigate the influence of intercropping on soil microbial communities, microbial activities and nutrient use efficiencies. The introduction is relevant and easy for readers to follow. However, the English language and the quality of scientific writing should be improved. Please recheck the spelling and grammar in the manuscript. Some numbers in the table and figure citations do not match the file provided.

Experimental design

The methods are generally appropriate, but more detail is required and the data analysis should be improved. I suggest that you improve the description of soil respiration and characterization of microbial community composition, and describe all statistical analysis used in this study (e.g. CCA, Spearman correlation were missing).

Validity of the findings

The interpretation of the results is insufficient and the conclusions are not fully justified by the results. For example, L188-190 & 244-245: why higher CUE/NUE ratios were due to higher respiration rates in intercropping treatments? higher respiration would lead to lower CUE as CUE can be calculated as Cgrowth/(Cgrowth+Crespiration) (e.g. Spohn et al. 2016; Keiblinger et al. 2014). are there any evidences that higher respiration could lead to even lower NUE? please elaborate.

Additional comments

L22-23: “the response of” and “in response to” are repetitive, rewrite the sentence.
L25: change “rhizospheric” to “rhizosphere”.
L28: define microbial activity (e.g., soil respiration).
L29: change ”effected” to “affected”.
L68: change “microbial structure” to “microbial community structure”.
L76: is it soil total N or soil total P?
L81: define “sieved soil” (<2mm?). change “considered” to “considered as”.
L87: add one sentence to describe how rhizosphere soil was recovered.
L91: add the company of the pH meter.
L95: remove “determined”.
L102-103: soil respiration rate is one of the key parameters measured in this study, briefly describe how it was measured instead of only referring to the literature.
L107&111: change “ITS1 genes” to “ITS1 region”.
L111: change “on” to “of”.
L117-121: move to “2.6. Statistical analysis”.
L126: Were relative abundances normal distributed here since you used ANOVA? Typically it is not easy to obtain normal distribution of microbial community data.
L134-135: wrong numbers when compared with Table 1.
L143-144: I suggest to calculate soil respiration rates based on soil dry mass or microbial biomass.
L146: wrong figure number, should be “Fig. 1b”.
L156&164: CCA and Spearman correlation were only mentioned in results part, but should be added to section 2.6.
L159-161: r values seem to be not correlated. are those p values?
L164: is Spearman correlation analysis based on mantel test?
L175: replace “such” with “such as”.
L180-181: the decrease of pH doesn’t necessarily indicate it was an important factor governing the microbial community. prove it (e.g. CCA, correlation).
L184-185: how did the SOC change microbial community? and why? please elaborate.
L191: soil total N and DON were not consistently higher in the intercropping treatments based on Table 1.
L199-200: only CUE/NUE ratios were calculated in this study, how could the author tell there was increased CUE? the ratio of fungi: bacteria (Sinsabaugh et al. 2013) rather than fungal abundance might shift soil microbial community CUE.
L214-215: The explanation here is not clear to me. what if dormancy rate of bacteria in soils is higher than that of fungi? then higher abundance of bacteria does not necessarily lead to higher microbial activity.
L217-218: TN and DON don’t seem to play important roles in shaping microbial community structure based on CCA and the results section.
L224-225: based on what kind of analysis? Fig. 4 can’t tell the abundances increased significantly.

Reviewer 2 ·

Basic reporting

The article compared the effect of intercropping and monoculture on soil microbial community composition and activities, the topic of the article is interesting and have values in soil managements. The structure of the article conforms to an acceptable format, and relevant literature references were appropriately provided. However, the statistical analysis should be improved. Some of the discussions are speculative and not supported by the data. Some sentences are just too long and need to be rephrased (such as line 218-220, 192-196).

Experimental design

Research question is not well defined, the objectives of the article are too ambiguous. More detailed hypotheses should be presented in the article, such as the specific difference between monoculture and intercropping on soil microbial communities. In addition, the ultimate goal of this study is to determine whether and how the intercropping management is better than monoculture for soil system, thus a conclusive statement is encouraged to be provided.
The experiment was designed scientifically, and the investigation was conducted to a high technical standard. However, some information was missing in the method section:
1, The procedure of collecting rhizosphere soil should be briefly described, since there are different methods for sampling (Barillot et al., Ann Microbiol, 2013, 471-476).
2, The test of CCA was conducted but it was not mentioned in the statistical analysis.
3, Least significant difference (LSD) was used in ANOVA, but pair-wise comparisons for 4 treatments (sugarcane monoculture; intercropped sugarcane; soybean monoculture; intercropped soybean) were not provided.

Validity of the findings

The article showed the valuable findings that intercropping changed soil physiochemical properties, and subsequently effected microbial structure and microbial activity. I have some questions and suggestions about the discussion and conclusion:
1, The discussion about the ratio of CUE and NUE was based on too much speculation rather than your own data. Please show your results, such as increased CUE or decreased NUE, and explain the results with caution.
2, Several environmental factors were analyzed in CCA, only the factors of pH and SOC were discussed, why?
3, The relative abundance of bacterial and fungal groups at the phylum, genus, and OTU levels were presented, but the analysis on microbial community was unsatisfactory. You specifically discussed some microorganisms, such as Pseudomonas, Fusarium, Chaetomium. Why did you pick these microorganisms? It has limitations to extrapolate to a larger scale by only describing the relationships between certain species and soil managements. To further explore the functional traits of microorganism at community levels in different treatments, I recommend you to use ‘Funguild’ for fungal OTUs to distinguish the pathogens (Nguyen et al., Fungal Ecol, 2015, 1-8).
4. The conclusions should be modified, they should be connected to the original question investigated; in addition, only one reference is published in 2018, please consider citing more latest references.

Additional comments

line22-24 Ambiguous sentence, please rephrase it.
Line 27-32 please show more specific results, with detailed description
line 45 ‘alters’ replace ‘shifts’
line 59 delete ‘were’
line 62-63 Italic for species names
line 65 ‘fungi’ to ‘fungal’
line 66-69 The objectives of this study are too ambiguous, please specify
line 87-88 Please briefly describe the procedure to collect rhizosphere soil
Figure 3 Please add the description for the cluster analysis

Reviewer 3 ·

Basic reporting

The manuscript is interesting and well written. Topic and question are relevant and timely. Lian and colleagues analyzed the bacterial and fungi structure and activity in the intercropping and monoculture system. They found that Intercropping changed soil physiochemical properties and microbial community structures. The novelty of this next-generation sequencing survey of soil microbial in the intercropping and monoculture system is the relationships between microbial community structures and microbial activity. As such, it has value.

Experimental design

The research questions are relevant and meaningful. The data analyses are pretty well explained. Below are some more specific comments that I hope are useful as the authors revise the manuscript.
Line 114-115: Did you remove the chimeric sequences from the row data? And how did you do that? Which chimera checking approach available in QIIME did you use?
Line 115-116: Which method did you choose to cluster the sequences into OTU? Such as CDhit, Uclust or some others? Did you standardise sequencing depth of each sample to a fixed value after get the OTUs? Such as 1000, 2000 or some other?
Line 120: please also report richness of microbial, not only Chao1.

Validity of the findings

Lian et al present a study which analyze simultaneously fungal and bacterial community structure in the intercropping and monoculture system. The research questions of the study certainly are current hot topics in microbial ecology. The study is meticulously well executed, the data is robust. The conclusion are well stated, which is linked to original research question and limited to supporting results.

---

## Round 0.2 · Minor Revisions

I do not think more experiments or data analysis are needed but the reviewer's comments should be addressed specifically.

Regards,

Michael

Reviewer 1 ·

Basic reporting

After one round revision, the manuscript has been improved a lot. However, there are still some concerns need to be addressed. Especially the description of results need to be more precise.

Experimental design

Line102-104: Are you sure SOC, DOC, DON were all measured by TOC analyser? Typically TC,TN were measured differently from DOC and DON measurements.
Line 132: Do you mean rarefy?
Line 137: Was PCoA based on OTU level?
Line 149: change "different on" to "differences of".

Validity of the findings

Some the of explanations are not satisfactory. See the detailed comments.

Additional comments

1. Change "physiochemical" to "physicochemical" across the manuscript.
2. Line 33: change "significant" to "significantly'.
3. Line 58: change"system have studied" to "systems have been studied".
4. Line 171-172: The description is not precise since bacterial and fungal abundances were only significantly increased for intercropped soybean.
5. Line 175-176: There were no figure captions for all supplementary figures, which makes the description in the MS confusing.
6. Figure 5a needs to be adjusted as one of the error bar was missing.
7.Line 201-202: Description is not precise. Plant pathogen did not increase significantly in I-Sugarcane.
8. Line 243-244: In this case, CUE could only decrease when respiration rates increased strongly than growth increase. Also, it is not clear to me why higher respiration led to lower ratios of carbon and nitrogen use efficiency.
9.Line 246-248: If no statistical significance was detected, I do not think this is a good explanation.
10.Line 258-259: Again since no statistical significance was detected in I-sugarcane, it is better to mention that bacterial and fungal abundances were only significantly higher in I-soybean system.
11. Line 261-263: The explanation is not satisfactory for me. As mentioned no statistical significance of available N was detected.
12. Line 263: "nutrient condition" are not proper words here.
13. Line 282-284: See comments 9 & 11.
14. Line 300: Change "microbial" to "bacterial".
15. Line 301: Change "was" to "as".
16. Line 304: Change "acceleration of" to "accelerate".

Reviewer 3 ·

Basic reporting

no comment

Experimental design

no comment

Validity of the findings

no comment

---

## Round 0.3 · Minor Revisions

One reviewer decided to accept and one declined to review the revised manuscript. I decided to accept the manuscript with minor changes. These changes are detailed in the attached pdf. Importantly, address the lack of discussion of Figure 1a.

---

## Round 0.4 · Minor Revisions

Importantly, please discuss Figure 1a, as suggested previously and make or address the changes below. If these issues are addressed, I do not see the need for further review.

Regards,

Michael

Line 51. Delete phrase “to maintain resource balances”
Line 56. Revise to “..communities have been studied in several intercropping systems, including mulberry..”
Line 61 – 70. Revise to “…and found that intercropping increased the abundance of some phosphate solubilizing species. Moreover, Rachid et al. ..changed the fungal structure and Zheng et al. (2018) reported that intercropping of apple trees and crown vetch changed the soil bacterial structure but not diversity.”
Line 73. Delete “Our results…monoculture.”
Line 93. Change to “May 25th, 2016”
Line 108. Move the sentence “The methods..(Lian et al. 2016) to start of paragraph.
Line 111. Fix my spelling error to “trapped”
Line 173. Revise to “..richness was only..”
Line 174. Change to “treatments”
Line 200. Revise to “with the ITS primer pair”
Line 223. Delete phrase “microbial properties of”
Line 216-217. Revise to “..system could relate to nitrogen fixation by the legume, which may..”
Line 223. Delete “Soil pH ..may be because” and revise to “Soil pH influences the acid-base..”
Line 226. Revise to “Additionally, CCA showed pH strongly..”
Line 242. Revise to “..abundance, without a change in available N, may explain the increase in carbon-use..”
Line 253. Revise to “..fungi are usually considered copiotrophic..”
Line 254. Pluralize “groups”
Line 268. Revise to “generalized that C mineralization predicts bacterial phyla abundances.”
Line 285. Delete “found”
Line 296. Revise to “..shifted soil microbial communities and soil physiochemical…”

---

## Round 0.5 · Minor Revisions

Please consider the following revisions.
Line 159-161: “Respiration peaked on day one and then decreased exponentially; respiration did not differ significantly between treatments (Fig. 1a).”

Line 211-218: “For soils collected from the intercropping or monoculture systems, nutrients and respiration decreased markedly from initial rates, which agrees with earlier studies (Lian et al., 2016) and the improvements in soil quality associated with intercropping appeared temporary.”

Line 299. Revise format to:
Bae SJ, Mohanta TK, Chung JY, Ryu M, Park G, Shim S. 2015. Trichoderma metabolites control phytopathogens of phytophthora blight. Biological Control 92: 128-138.

Line 327-331 and 388-395. Two references are concatenated.

Regards,

Michael

---

## Round 0.6 · accepted · Accept

I appreciate you patience and confirm that this article is now Acceptable

Regards,

Michael